# Epithelial-specific loss of Smad4 alleviates the fibrotic response in an acute colitis mouse model

Zahra Hashemi[1], Thompson Hui[1], Alex Wu[2], Dahlia Matouba[1], Steven Zukowski[1], Shima Nejati[1], Crystal Lim[1], Julianna Bruzzese[1], Cindy Lin[1], Kyle Seabold[1], Connor Mills[1], Kylee Wrath[1], Haoyu Wang[1], Hongjun Wang[1], Michael P Verzi[2], Ansu Perekatt[1]

**Mucosal healing is associated with better clinical outcomes in patients with inflammatory bowel disease. But the epithelial-specific contribution to mucosal healing in vivo is poorly understood. We evaluated mucosal healing in an acute dextran sulfate sodium mouse model that shows an alleviated colitis response after epithelial-specific loss of Smad4. We find that enhanced epithelial wound healing alleviates the fibrotic response. Dextran sulfate sodium caused increased mesenchymal collagen deposition—indicative of fibrosis—within a week in the WT but not in the Smad4 KO colon. The fibrotic response correlated with decreased epithelial proliferation in the WT, whereas uninterrupted proliferation and an expanded zone of proliferation were observed in the Smad4 KO colon epithelium. Furthermore, the Smad4 KO colon showed epithelial extracellular matrix alterations that promote epithelial regeneration. Our data suggest that epithelium is a key determinant of the mucosal healing response in vivo, implicating mucosal healing as a strategy against fibrosis in inflammatory bowel disease patients.**

## Introduction

Mucosal healing is a prime treatment goal for IBD patients (1, 2). As one of the most extensive epithelial linings in the body (3), the intestinal epithelium provides a critical barrier between the microflora in the intestinal lumen and the gut-associated immune cells (4). Hence, rapid resealing of the epithelial barrier after epithelial injury is essential to restore homeostasis and to prevent inflammation (5, 6).

When the regenerative capacity of the epithelia is compromised or does not suffice to meet the extent of injury, exposure of the luminal contents to the underlying mesenchyme triggers an inflammatory response (7). When unchecked, the inflammatory response can lead to fibrosis, which can lead to strictures, a common complication in Crohn's disease (CD) (8). Chronic inflammation can also increase the predisposition to cancer in ulcerative colitis (UC) and CD patients (9).

The DSS mouse model is a well-established animal model to study acute and chronic inflammatory responses in the intestine. DSS administration in drinking water is thought to cause injury by penetrating the epithelial membrane (10); the severity of the injury and inflammation depends on the molecular weight of DSS, the amount administered in the drinking water and the treatment period (11). A restorative response to DSS-induced injury involves several steps: hemostasis to seal breached vasculature, cell cycle arrest, spreading of the epithelial cells to cover the denuded area, and differentiation to restore the epithelial integrity (12). Parallelly, the immune cells are activated to clear the infiltrated immunogens (13). However, an unresolved injury because of compromised regenerative capacity can cause fibrosis, a pathological wound-healing response wherein fibroblasts and collagen replace the epithelium (14).

The healing process in chronic wounds causes systemic changes that desensitize the immune cells, thereby creating an immunosuppressive milieu that permits the growth of neoplastic cells (14, 15). Loss-of-function mutation(s) in Smad4, a tumor suppressor (16, 17, 18, 19), is(are) one of the critical drivers of colon cancers (20). Although Smad4 loss alone in the intestinal epithelium does not affect the gross phenotype (21, 22), it increases colitis-associated cancer in the chronic DSS mouse model (23, 24).

We used the acute DSS model to delineate the early wound-healing response in the epithelium without the confounding effects of chronic inflammation. Smad4 was knocked out specifically in the intestinal epithelium (Smad4^IEC–KO), followed by DSS treatment. Epithelial-specific transcriptional and molecular changes were assessed after 3 d of 2.5% DSS (3 d post-DSS). We chose the 3-d time point as the DSS-induced loss of the colonic epithelial tissue is minimal at this time point—enabling the collection of the sufficient quality of the epithelial tissue for transcriptomic analysis at a time point when early molecular responses are evident (25). Because the gross phenotypic change manifests within 7 d of continuous DSS treatment in an acute colitis model (18), we chose the 7-d time point to evaluate the gross phenotypic effects of DSS after 7 d of 2.5% DSS (7 d post-DSS).

[1]Department of Chemistry and Chemical Biology, Stevens Institute of Technology, Hoboken, NJ, USA   [2]Department of Genetics, Rutgers University, Piscataway, NJ, USA

Correspondence: aperekat@stevens.edu

We find that the epithelial-specific loss of Smad4 has a protective effect against the colitis response in the acute DSS model. Most interestingly, Smad4 loss alleviated the fibrotic response, which is characterized by the expansion of the mesenchymal tissue and collagen deposition in the mesenchyme [15, 16]. Our findings reveal epithelial-specific ECM changes supporting epithelial regeneration after DSS in the Smad4$^{IEC-KO}$ mice.

# Results

## Epithelial-specific loss of Smad4 in the acute DSS mouse model alleviates the colitis response

To determine the effect of epithelial-specific loss of Smad4 on colitis response in an acute DSS mouse model, we first knocked out Smad4 in the epithelium (Smad4$^{IEC-KO}$) using the Villin promoter–driven, tamoxifen-inducible, Cre-recombinase [17] (Fig S1B) and assessed the colitis response to 2.5% DSS (40 kD) in drinking water for 7 d. DSS caused weight loss and reduction in colon length in both the WT and the Smad4$^{IEC-KO}$ mice. However, the colitis-associated phenotype was significantly alleviated in the Smad4$^{IEC-KO}$ mice (Fig 1A and B). Furthermore, histological evaluation by hematoxylin and eosin (H&E) and quantification revealed reduced tissue and crypt damage in the 7-d post-DSS Smad4$^{IEC-KO}$ mouse colon (Fig 1C and D). These observations collectively indicated a protective effect of Smad4 loss in the intestinal epithelium against DSS in the acute DSS mouse model of colitis.

## Epithelial-specific loss of Smad4 promotes the regenerative response in the epithelium

DSS triggers inflammation by causing epithelial breaches that expose the colonic luminal contents to the underlying mesenchyme [18, 19]. Hence, faster resealing of the breached epithelium can minimize the DSS-induced inflammation and colitis. We first assessed epithelial proliferation and migration, which are wound-healing responses that reseal the breached epithelium to cover the denuded areas [20, 21]. DSS caused a significant reduction in the epithelial proliferation in the WT colon within a week, but not in the Smad4$^{IEC-KO}$ colon epithelium, suggesting the lack of the DSS-induced proliferative arrest [22] in the Smad4$^{IEC-KO}$ colon (Fig 2A and B).

We then assessed epithelial migration because we observed an extended proliferative zone in the Smad4$^{IEC-KO}$ colon epithelium after DSS (Fig S2D). To this end, we performed EdU (5-ethynyl-2'-deoxyuridine) and BrdU (bromodeoxyuridine) dual pulse-chase assays for one and 6 h, respectively. First, we measured the distance at which the top-most EdU-labeled and BrdU-labeled cells were found relative to the crypt height, consistent with the observations for Ki67 immunostaining (Fig S2F). The EdU-labeled and BrdU-labeled cells were significantly higher up in the crypts of the Smad4$^{IEC-KO}$ colon after 3 d of DSS (Fig 2C–E). However, when the distance between the top-most

EdU- and the top-most BrdU-positive cells within the same crypt was quantified, no significant difference between the 3-d DSS WT and the 3-d DSS Smad4$^{IEC-KO}$ could be observed (Fig S2E), indicating an expansion in the proliferative zone rather than increased epithelial migration.

To further evaluate the early epithelial-specific transcriptional changes that support the regenerative response, we performed RNA-seq on the colonic epithelium of the 3-d DSS-treated mice, followed by gene set enrichment analysis (GSEA). The colitis-associated epithelial regenerative gene signature [23] was significantly enriched in the Smad4$^{IEC-KO}$ epithelium compared with WT both before and after 3 d of DSS (Fig 2F), indicating epithelial-specific transcriptional changes that promote epithelial regeneration in the Smad4$^{IEC-KO}$ colon.

Next, to investigate homeostasis restorative responses, we probed for differentiation markers. The Smad4 KO epithelium showed the increased expression of keratin 20 (Fig 2G) and E-cadherin (Fig 2H) within 3 d of DSS, suggesting DSS-induced responses that restore differentiation [24, 26] and epithelial integrity [27, 28] in the Smad4$^{IEC-KO}$. These data collectively indicate enhanced epithelial wound healing and homeostasis restorative responses to DSS in the Smad4$^{IEC-KO}$ colon.

## Epithelial-specific loss of Smad4 alleviates the fibrotic response in the acute DSS mouse model

As the number of intact epithelial crypts remaining 7 d post-DSS was significantly higher in the Smad4$^{IEC-KO}$ colon compared with its WT counterpart (Fig 1C and D), we assessed the pathological wound-healing response characterized by replacement of the damaged tissue with the non-epithelial stromal tissue [29, 30]. Increased fibroblast activity and deposition of collagen-containing ECM components in the stroma during pathological wound healing lead to fibrosis [31, 32, 33]. To determine the fibrotic response, we first assessed collagen deposition using picrosirius red staining (Fig 3A) and quantified the collagen proportionate area [34] in the mucosa. Collagen proportionate area increased significantly after 7 d of DSS in the WT. However, the collagen proportionate area was unaffected by DSS in the Smad4$^{IEC-KO}$ colon (Fig 3B), suggesting epithelial-specific Smad4 loss protects against the fibrotic response in the acute DSS mouse model.

We next probed for α-SMA (α-smooth muscle actin), a marker of activated fibroblasts that promotes collagen deposition [35] in the stroma [36, 37] and fibrosis. Immunoreactivity for α-SMA was evident 7 d post-DSS in the regions lacking intact crypts in both the WT and Smad4$^{IEC-KO}$ colon (Fig S3G). However, the stroma-only region, that is, a mucosal region without intact crypts, was significantly lower in the 7-d post-DSS Smad4$^{IEC-KO}$ colon when compared to its WT counterpart (Fig 3D), suggesting increased fibroblast activity in the stroma of the WT of the colon after DSS. Surprisingly, the Smad4$^{IEC-KO}$ colon showed stronger α-SMA immunoreactivity in the pericryptal region lining of the crypt epithelium when compared to its WT counterpart even before DSS treatment (Figs 3C and S3H), suggesting altered crosstalk between the epithelium and pericryptal fibroblasts after epithelial-specific Smad4 loss [36, 38].

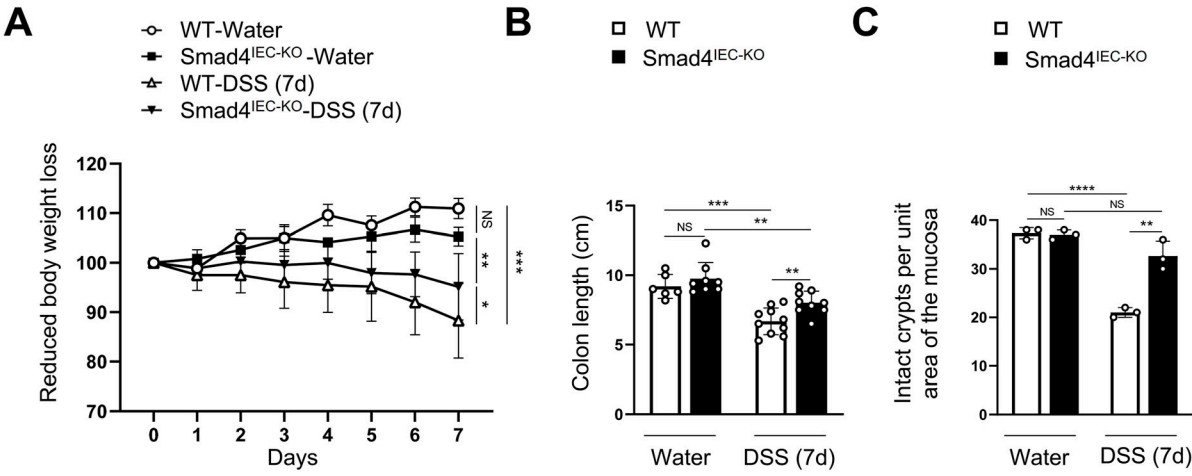

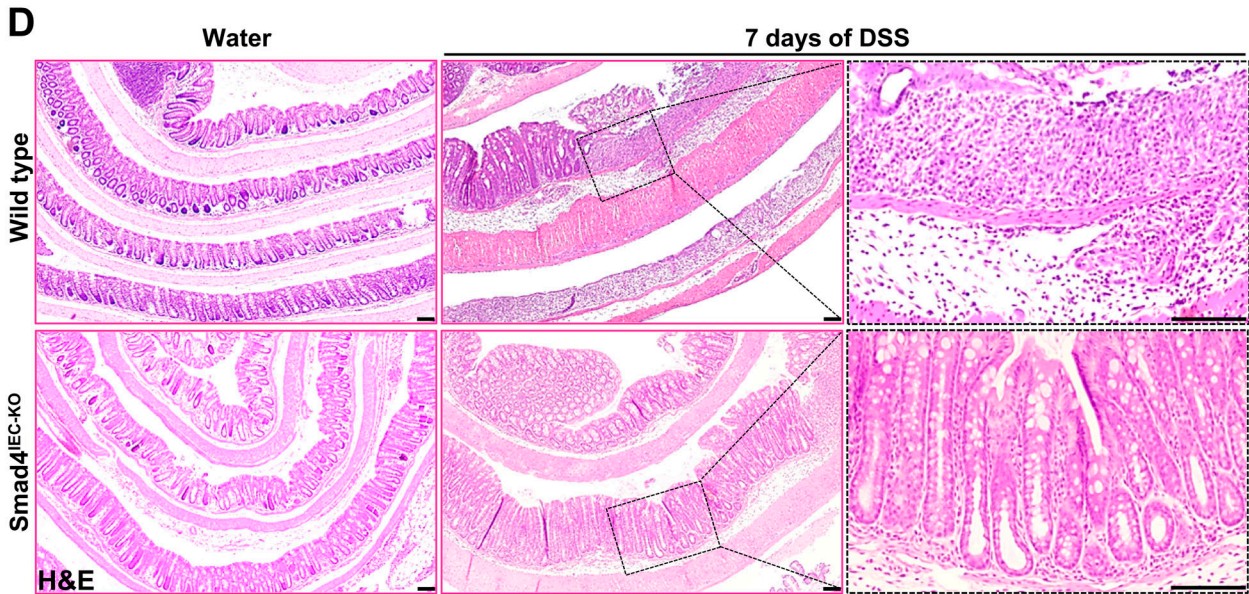

**Figure 1. Smad4 loss in the epithelium alleviates the colitis response in the acute DSS mouse model.**
**(A)** Loss in bodyweight (n = 35: 3 replicates/water-treated group, and 12 and 17 replicates for the WT and the Smad4^IEC–KO mice treated with 2.5% DSS for a week). **(B)** Reduction in colon length after a week of 2.5% DSS (n = 33: 6 replicates/WT, 8 replicates/Smad4^IEC–KO, 10 replicates/7-d post-DSS WT, and 9 replicates/7-d post-DSS Smad4^IEC–KO). **(C)** Quantification of intact crypts after 7 d of 2.5% DSS (n = 3 replicates/group). **(D)** Representative images of the pathological histology, determined by hematoxylin and eosin (n = 3 replicates/group). Scale bars, 100 μm. **(C, D)** All experiments in (C, D) were n = 3 independent experiments. In (A, B, C), data are presented as the mean ± SEM. Statistical significance was determined using an unpaired two-tailed $t$ test. NS, not significant. *, **, ***, and **** denote $P < 0.05$, $P < 0.01$, $P < 0.001$, and $P < 0.0001$, respectively.

## The epithelial transcriptome in the 3-d DSS-treated Smad4^IEC−KO colon supports the wound-healing response

To investigate the early epithelial-specific transcriptional response to DSS, we performed mRNA sequencing on the colon epithelia from the WT and the Smad4^IEC–KO mice. We used the 3-d post-DSS time point as the molecular changes, without affecting the gross epithelial morphology, are evident within 3 d of DSS (18, 39). ECM organization, collagen components, and wound healing were among the top-enriched GO (Gene Ontology) terms in the 3-d post–DSS-treated Smad4 KO colon epithelium compared with its WT counterpart (Fig 4A). Notably, the "ECM constituent" gene signature was significantly enriched in the Smad4 KO colon epithelium, when compared to its WT counterpart within 3 d of DSS (Fig 4B). Several of the genes in the ECM constituent signature, such as *Ltbp1* (40), *Prg4* (41), and *Dcn* (42), have been implicated in wound healing. Interestingly, members of small leucine-rich repeat proteoglycans (SLRPs), which are down-regulated during pathological

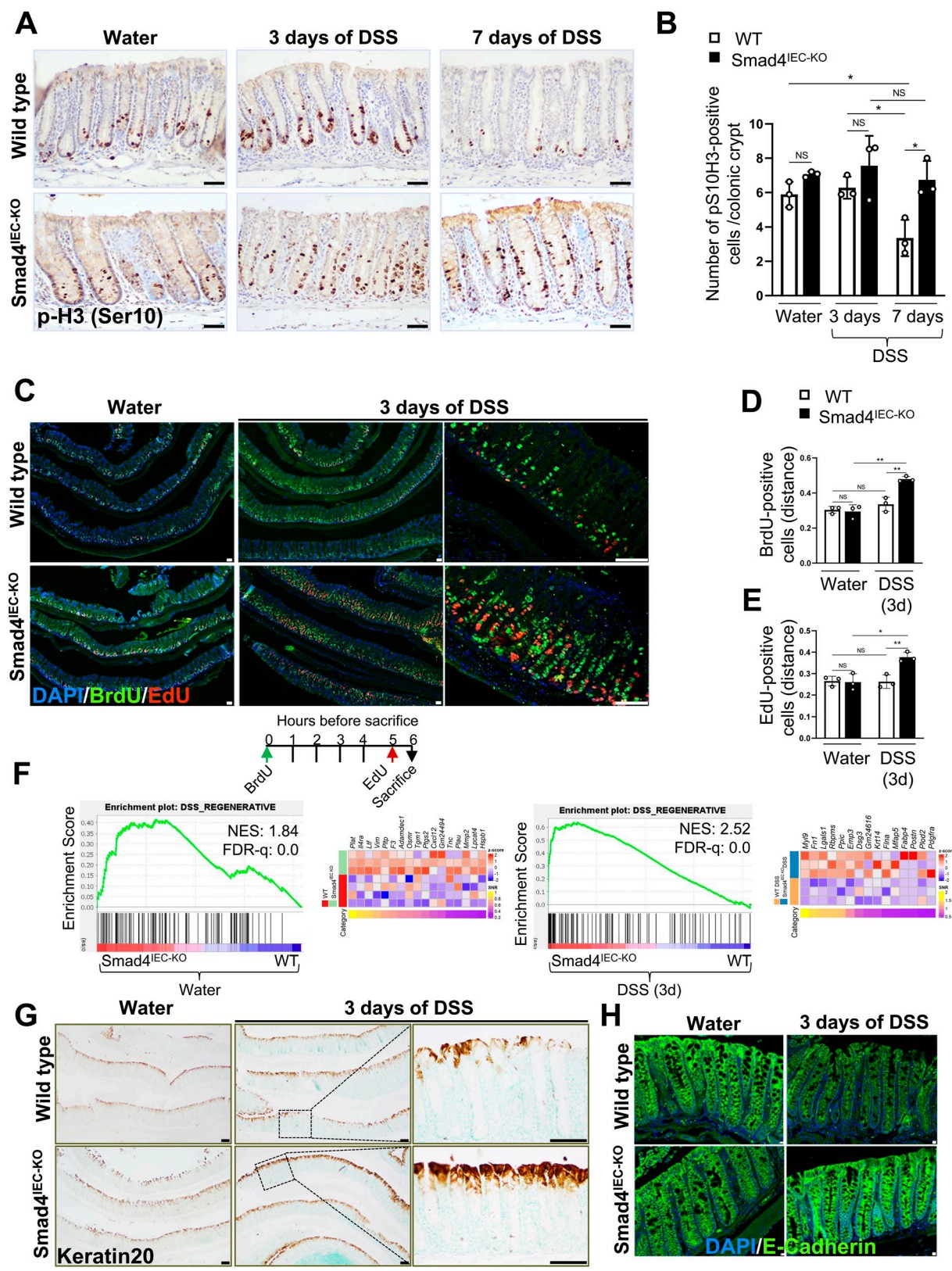

**Figure 2. Enhanced regenerative response in the Smad4 KO epithelium within 3 d of 2.5% DSS.**
**(A, B)** Representative images and (B) quantification of the proliferative response to DSS, determined by IHC for the mitotic marker p-H3 (Ser10). **(C, D, E)** Representative images of the expanded zone of proliferation, determined by the EdU and BrdU dual pulse-chase assay, and the distance relative to the crypt height at which (D) BrdU-

progression of colitis (SLRPs) (43), were positively enriched in the ECM signature, suggesting protective effects of the Smad4[IEC−KO] ECM constituents. In addition, the transcript levels of genes encoding the collagens implicated in wound healing (44) such as types I, II, IV, and VI (29) were higher in the Smad4 KO colon epithelium (Fig 4C). Because collagen in the ECM is a regulator of the various processes involved in epithelial regeneration and homeostasis (45, 46, 47), we probed for Col1a1, a type I collagen, by immunostaining (Fig 4D). Type I collagen increased significantly in the Smad4[IEC−KO] peri-cryptal epithelium after 3 d of DSS (Fig 4E), suggesting increased collagen in the epithelial ECM. To further evaluate the functional implication of the ECM-related gene signatures, specifically in the epithelium, we performed Ingenuity Pathway Analysis (IPA) on the differentially expressed genes (DEGs) in the 3-d post-DSS Smad4[IEC−KO] colon epithelium versus its WT counterpart. Collagen-mediated glycoprotein 6 (Gp6) signaling, which through platelet aggregation (48) resolves vascular breaches, was among the most enriched signaling pathways in the DSS-treated Smad4[IEC−KO] colon (Fig S3A and C). No difference in Gp6 expression was observed, however (Fig S3D). These observations in the Smad4[IEC−KO] colon implicate the increased epithelial ECM collagen in mucosal healing.

### Epithelial-specific loss of Smad4 attenuates the DSS-induced inflammatory response in the acute DSS mouse model

After DSS-induced damage, the pattern recognition receptors in the epithelium trigger inflammatory signaling (49). Hence, we first determined the transcriptional changes affecting the DSS-induced inflammatory response. The proinflammatory IFN-$\alpha$ and IFN-$\gamma$ (50, 51) gene signatures were enriched in the WT colon epithelium after 3 d of DSS (Fig 5A) but not in the Smad4 KO epithelium (Fig S1C). Consistent with this, the IFN-$\alpha$ and IFN-$\gamma$ gene signatures were negatively enriched in the 3-d post-DSS Smad4 KO epithelium compared with its DSS-treated WT counterpart (Fig 5B), indicating the attenuated DSS-induced inflammatory response in the epithelium.

To investigate DSS-induced immune cell infiltration because of Smad4 loss in the epithelium, we probed for iNOS (inducible nitric oxide synthase) and CD206, the pro- and anti-inflammatory markers of the innate immune response (52, 53). Decreased iNOS immunoreactivity was detected in the Smad4[IEC−KO] colon 7 d post-DSS compared with its WT counterpart, indicating a subdued proinflammatory response to DSS in the Smad4[IEC−KO] colon (Fig 5C). Conversely, immunoreactivity for the anti-inflammatory M2 macrophage CD206 was higher in the 7-d post-DSS Smad4[IEC−KO] colon compared with that in the WT colon (Fig 5D), suggesting suppression of proinflammatory responses in the Smad4[IEC−KO] colon after DSS treatment. To determine whether the pro- and anti-inflammatory changes correlated with the level of inflammation, we next assayed

the C-reactive protein (CRP), an indicator of inflammation (54). A significant increase in the CRP level was evident in the WT colonic tissue after 7 d of DSS but not in the Smad4[IEC−KO] (Fig 5E), indicating a dampened inflammatory response in the Smad4[IEC−KO] colon. These data collectively suggest that epithelial-specific loss of Smad4 attenuates the inflammatory response in the acute DSS mouse model.

## Discussion

Here, we show that epithelial-specific loss of Smad4 enhances epithelial regeneration, effectively alleviating the fibrotic response in the acute DSS mouse model. The regenerative response, which includes DSS-induced expansion of the proliferative zone, paralleled epithelial retention and reduced accumulation of the mesenchymal tissue in the Smad4[IEC−KO] acute DSS model. The transcriptional profile of the Smad4 KO epithelium also suggested epithelial-specific ECM alterations that support wound healing in the acute DSS mouse model.

Our studies are distinct from the previously reported tumorigenic effect of Smad4 loss in mouse models of chronic inflammation (55, 56, 57). Previous studies used haploinsufficiency (58) or partial deletion (55) of Smad4 in the epithelium and showed that Smad4 loss promotes colitis and colitis-associated cancer (55, 57). However, given the role of Smad4 in genomic stability (59, 60, 61) and tumor suppression (62, 63), tumorigenesis in the Smad4 KO chronic DSS mouse model is not surprising, especially in the presence of a DNA-damaging agent such as AOM (azoxymethane) (55, 56, 57). Therefore, we expect tumorigenesis in the DSS-treated Smad4[IEC−KO] colon, especially considering the immunosuppressive milieu (Fig 5C and D) if a long-term DSS regimen for chronic colitis was adopted. A striking observation in the Smad4[IEC−KO] colon was the DSS-induced wound-healing response. A key factor ascribed to DSS-induced colitis is the cell cycle arrest (22). Consistent with this, the WT of the colon displayed a significant decrease in epithelial proliferation within 7 d of DSS in Fig 2A and B. However, DSS did not decrease epithelial proliferation in the Smad4[IEC−KO] colon (Fig 2B). Furthermore, the transcriptional profile of the 3-d post-DSS Smad4 KO epithelium showed negative and positive enrichment for the GO-term "Cell Cycle Arrest" (Fig S2A) and the pro-proliferative "Myc target" gene signature (64) (Fig S2C), respectively. The decreased Cdkn1a transcripts encoding the cell cycle inhibitor p21 (Fig S2B) further indicated the lack of proliferative arrest in the Smad4[IEC−KO] colon epithelium. The DSS-induced expansion of the proliferative zone in the Smad4 KO epithelium is consistent with the epithelial injury responses that support wound healing (47). In addition, Lgr5 transcript levels were significantly higher in the DSS-treated Smad4[IEC−KO] colon epithelium (Fig S2H), consistent with the role

---

positive and (E) EdU-positive cells were found after the chase; the schematic depicts the timeline of BrdU and EdU pulse-chase (n = 3 replicates/group; scale bars, 50 $\mu$m). **(F)** Enrichment of the regenerative gene signature in the Smad4[IEC−KO] colon epithelium compared with its WT counterpart before and after 3 d of 2.5% DSS, visualized by the GSEA plot and the corresponding heatmap of the core-enriched genes (n = 4 replicates/water-treated WT and 3 replicates/other groups). **(G, H)** Representative images of improved epithelial differentiation and integrity, determined by (G) keratin 20 (n = 3 replicates/group; scale bars, 100 $\mu$m) and (H) E-cadherin immunostaining (n = 3 replicates/group; scale bars, 10 $\mu$m). **(A, B, C, D, E, G, H)** All experiments in (A, B, C, D, E, G, H) were n = 3 independent experiments. In (B, D, E), data are presented as the mean ± SEM. Statistical significance was determined using an unpaired two-tailed $t$ test. NS, not significant. * and ** denote $P < 0.05$ and $P < 0.01$, respectively.

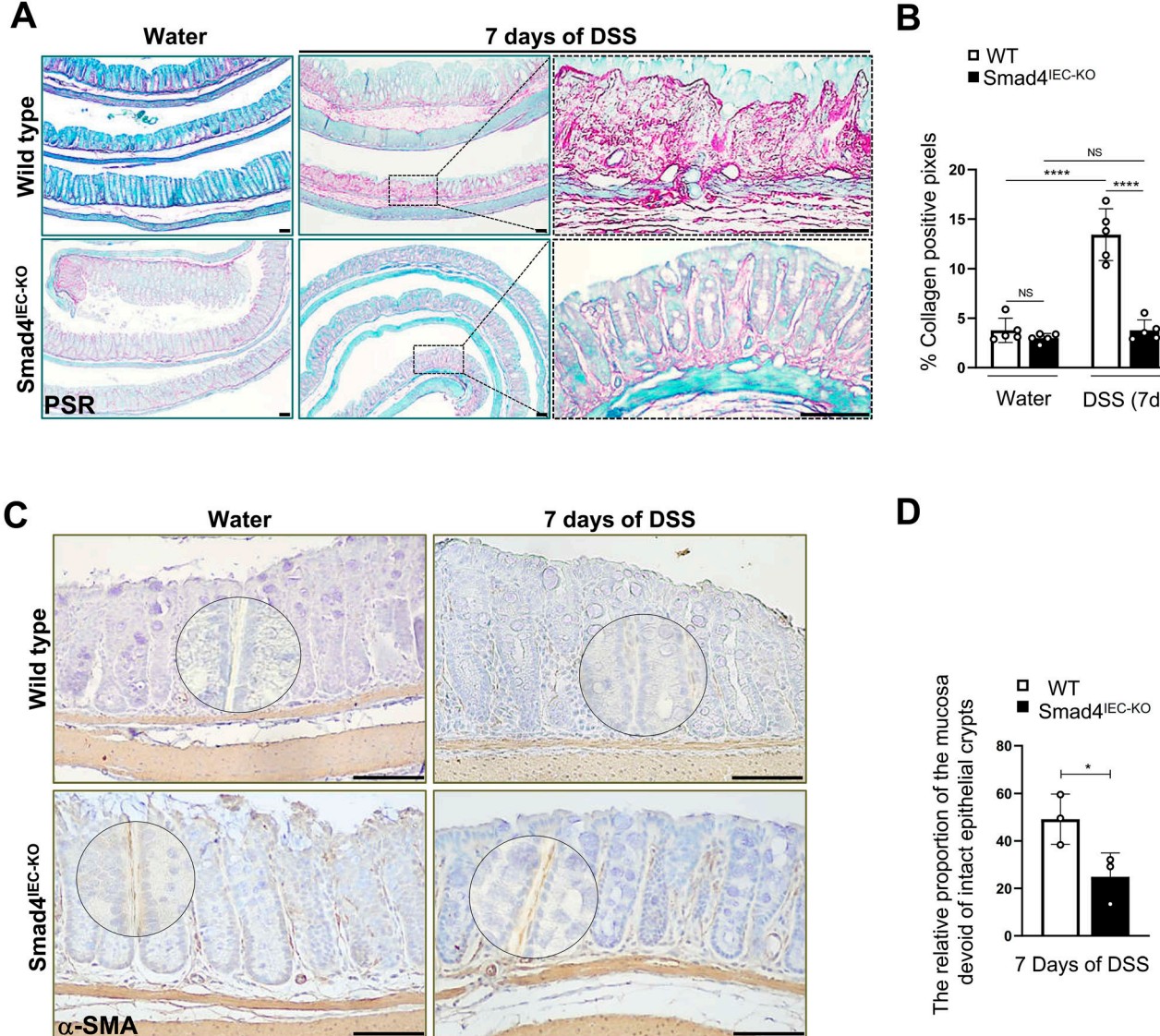

**Figure 3. Reduced fibrotic response in the acute DSS mouse model with epithelial-specific Smad4 deletion.**
**(A, B)** Representative images of collagen deposition, determined by picrosirius red staining, and (B) collagen proportionate area, determined by the percentage of collagen-positive pixels, after a week of 2.5% DSS. **(C)** Representative images of the increased fibroblast activity in the pericryptal fibroblast, determined by $\alpha$-SMA-IHC; the encircled are the magnified regions. **(D)** Relative proportion of the $\alpha$-SMA–positive areas in the mucosal regions devoid of intact crypts (n = 3 replicates/group). Scale bars, 100 $\mu$m. All experiments in this figure were n = 3 independent experiments. In (B, D), data are presented as the mean ± SEM. Statistical significance was determined using an unpaired two-tailed $t$ test. NS, not significant. * and **** denote $P < 0.05$ and $P < 0.0001$, respectively.

of Lgr5+ stem cells in the intestinal epithelial regeneration (65, 66). Thus, the homeostatic responses and the enrichment of the regenerative gene signatures (23) (Fig 2F) in the epithelium suggest that Smad4 loss has a protective effect against the DSS-induced colitis response.

Given that Smad4 is a transcriptional effector of TGF-$\beta$ signaling, our finding that Smad4 loss promotes epithelial regeneration is intriguing. TGF-$\beta$ promotes epithelial regeneration after ionizing radiation (IR) through fetal reprogramming (67) and by suppressing inflammation (68). However, the inflammatory response is minimal after IR compared with DSS. The subdued inflammatory response after IR is attributed to the phagocytic clearance of the apoptotic

cells (69, 70). On the contrary, the DSS-induced epithelial breach exposes the luminal contents to the epithelium, thereby engaging damage-associated molecular patterns on the epithelium (71), which might trigger various lytic forms of cell death, which in turn might prolong the inflammatory response.

Although the Smad4$^{IEC-KO}$ colon showed no increase in mesenchymal collagen deposition in the mucosa after DSS (Fig 3A and B), the pericryptal collagen deposition was higher (Fig 4D) in the Smad4$^{IEC-KO}$ colon. This is consistent with the context-dependent functions of collagens. The fibrosis versus the wound-healing attributes of various collagens depend on the source of collagen and the site of its deposition. For example, fibroblast deposition of type I

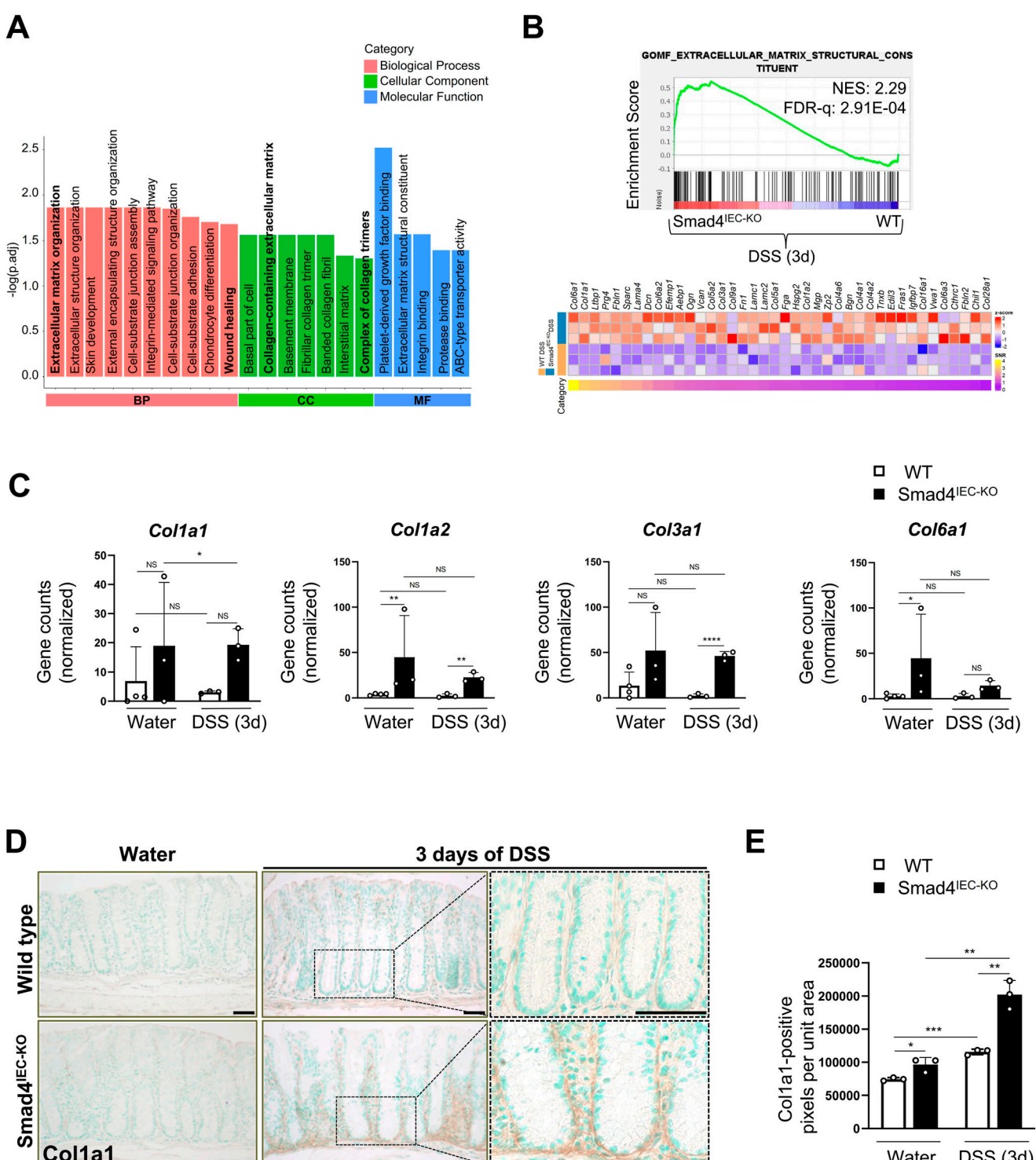

**Figure 4. Epithelial-specific ECM alterations reflective of wound-healing responses in the Smad4[IEC-KO] colon within 3 d of 2.5% DSS.**
**(A)** ECM-related changes and wound healing among the top GO terms enriched in the 3-d post-DSS Smad4[IEC-KO] epithelial transcriptome compared with its WT counterpart (n = 3 replicates/group). **(B)** Gene signature enrichment of the ECM component function in the Smad4[IEC-KO], visualized by the GSEA plot and heatmap of the core-enriched genes (n = 3 replicates/group). **(C)** Transcriptional increase in the genes encoding the collagens, Col1a1, Col1a2, Col3a1, and Col6a1, implicated in the wound-healing response (n = 4 replicates/water-treated WT and 3 replicates/other groups). **(D, E)** Representative images of increased type I collagen in the Smad4[IEC-KO] epithelial ECM, determined by Col1a1 IHC, and (E) quantification of the Col1a1 immunoreactive pixels in the epithelial ECM (n = 3 replicates/group). Scale bars, 50 μm. All experiments in (D, E) were n = 3 independent experiments. In (C), the statistical significance is based on Padj. values from the DEG analysis. In (E), data are presented as the mean ± SEM. Statistical significance was determined using an unpaired two-tailed t test. NS, not significant. *, **, ***, and **** denote P < 0.05, P < 0.01, P < 0.001, and P < 0.0001, respectively. NES = normalized enrichment score, FDR-q = false discovery rate q-value.

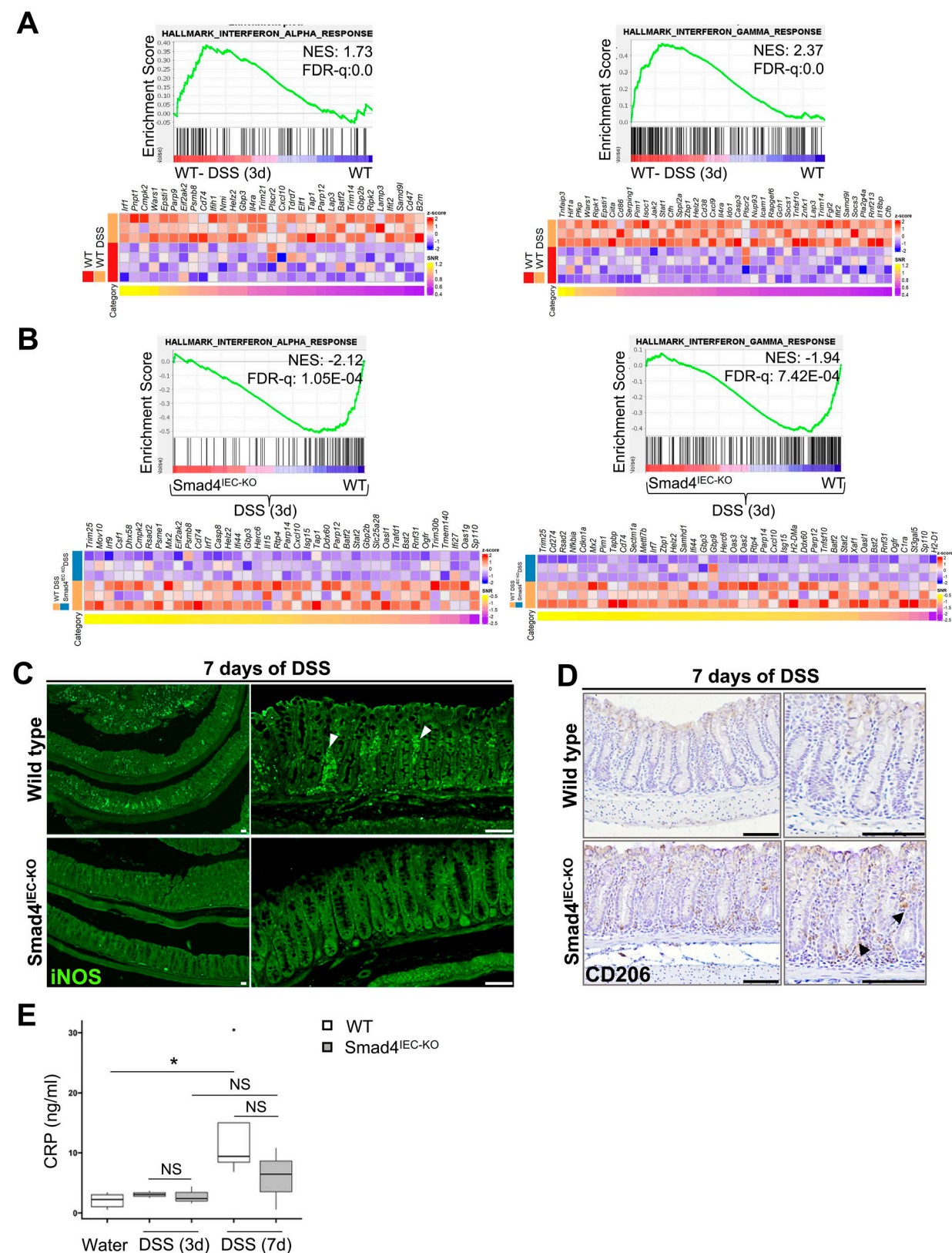

**Figure 5. Attenuated inflammatory response to DSS in the Smad4[IEC-KO] colon.**
**(A)** Positive enrichment of the proinflammatory IFN-α and IFN-γ gene signatures after 3 d of 2.5% DSS in the WT colon epithelium, visualized by the GSEA plot and the corresponding heatmap (n = 4 replicates/water-treated WT and 3 replicates/3-d post-DSS WT). **(B)** Negative enrichment of the proinflammatory IFN-α and IFN-γ gene

and III collagens in the mesenchyme is attributed to fibrosis (72), whereas type I and III collagens, when present in the epithelial ECM, are associated with wound healing (73). Likewise, although type VI collagen in the epithelial basement membrane is associated with wound-healing processes (44, 74), an increase in type VI collagen has been reported in the strictures of CD, and in collagenous colitis (75). Type I collagen in the epithelial ECM promotes epithelial proliferation (46), migration, and differentiation even when the cell–cell contact is absent (45, 76). Because the denuded epithelium resembles epithelial cells lacking the cell–cell contact, the increased type I collagen in the pericryptal region (Fig 4D and E) might restore faster homeostasis. In addition, the increase in the pericryptal collagen might promote hemostasis by serving as a ligand for the platelet-activating Gp6 signaling (77, 78) (Fig S3B). Although no increase in Gp6 immunoreactivity was detected (Fig S3D), the increased pericryptal collagen in the Smad4$^{IEC-KO}$ colon (Fig 4D and E) might contribute to the wound-healing response (Fig S3F) through Gp6 signaling (Fig S3A and C) by resealing the injured vasculature (79).

Epithelial and the pericryptal fibroblast crosstalk can increase collagen deposition and remodeling of the epithelial ECM (80). Thus, the increased pericryptal collagen in the Smad4$^{IEC-KO}$ colon (Fig 4D and E) could also be ascribed to the pericryptal fibroblast activity (36, 38) (Figs 3C and S3H). This notion is consistent with the higher expression of platelet-derived growth factor alpha (Fig S3E), which encodes the ligand for the fibroblast receptor, platelet-derived growth factor receptor alpha (81, 82). However, additional studies are needed to understand the basis for the differential activation of the pericryptal fibroblasts and their contribution to epithelial homeostasis in the acute DSS model.

Although the alleviated colitis response could be ascribed in part to the epithelial restorative responses after injury, the dampened inflammatory response (Fig 5B, C, and E) and infiltration of the anti-inflammatory M2 macrophages (83) after DSS (Fig 5D) implicate epithelial-specific Smad4 loss in immunomodulation and suppression of the pathological inflammatory response. This observation warrants further investigation, especially because anti-inflammatory immune infiltration negatively regulates fibrosis (84). Our study underscores the importance of epithelial restoration in alleviating the fibrotic response to injury in the acute DSS mouse model. In this regard, the differential role of the pericryptal and the mesenchymal fibroblasts in epithelial restoration versus fibrosis warrants further investigation. In conclusion, our study reveals that the enhanced regenerative response in the Smad4$^{IEC-KO}$ is associated with epithelial–ECM changes and alleviated fibrosis. Hence, exploiting the epithelial–ECM changes that promote epithelial regeneration is a potential strategy against fibrosis in IBDs.

# Materials and Methods

## Materials

Antibodies and reagents are summarized in Tables S1, S2, and S3.

## Animals and experimental protocol for inducing colitis

The animal experiments were conducted according to the protocol approved by the IACUC of Stevens Institute of Technology and Rutgers. All mice were kept under a 12-h light/dark cycle and harvested around midday to prevent diurnal variations. To create the Smad4 KO conditional mutant, the Villin-Cre$^{ERT2}$ transgene (17) was integrated into Smad4$^{f/f}$ (85) mouse conditional mutants and controls. To induce epithelial-specific loss of Smad4, 0.05 g/kg tamoxifen per day was injected intraperitoneally for four consecutive days. 2.5% DSS (40 kD) was administered in drinking water for ad libitum drinking for three or seven consecutive days, depending on the purpose of the experiment. The treatment regimen for Smad4 deletion and DSS treatment is shown (Fig S1A). The animals used were gender- and age-matched across the different treatments. All animals used were between 8 and 14 wk of age.

## Colon epithelial isolation for RNA extraction and sequencing

The freshly harvested colon was flushed with PBS, filleted open, and cut into ~1-cm pieces, and the epithelia were separated from the underlying mesenchyme using EDTA chelation as follows: the intestinal pieces were incubated with 5 mM/liter EDTA/PBS for 50 min at 4°C and then shaken to separate the epithelium from the underlying mesenchyme. The colonic crypt epithelium was isolated after filtering through a 70-$\mu$m filter. The isolated crypts were washed twice with ice-cold PBS by spinning at 200 rcf at 4°C for 2 min. Any remaining PBS was aspirated before solubilizing in TRIzol and RNA extracted as per the manufacturer's recommendation.

## Protein extraction

The whole colonic tissue from the proximal colon was used for protein extraction. After flushing the colon with cold PBS, the proximal colon was minced into 1-cm pieces and flash-frozen in liquid nitrogen and macerated using a pestle and mortar. The macerated tissue was lysed in RIPA buffer (20 mM/liter Hepes, 150 mM/liter NaCl, 1 mM/liter EGTA (ethylene glycol-bis[$\beta$-aminoethyl ether]-N, N,N′,N′-tetraacetic acid), 1% Triton X-100, and 1 mM/liter EDTA) containing freshly added protease inhibitors (1x PI, 20 mM/liter NaF, 1 mM/liter Na3VO3, and 1 mM/liter PMSF), rocked at 4°C for 30′, and spun at 4°C to extract the solubilized protein. The protein

---

signatures in the 3-d post-DSS Smad4$^{IEC-KO}$ epithelium compared with its WT counterpart, visualized by the GSEA plot; the corresponding heatmap shows the relative levels of the core-enriched genes (n = 3 replicates/group). **(C, D)** Representative images of the decreased proinflammatory response after 7 d of 2.5% DSS in the Smad4 IEC-KO colon, determined by immunostaining for (C) the proinflammatory marker, iNOS (white arrowheads), and (D) the anti-inflammatory M2 macrophage marker, CD206 (black arrowheads) (n = 3 replicates/group). **(E)** Increased inflammation in the WT but not the Smad4$^{IEC-KO}$ colon, determined by CRP levels in colonic tissues after 7 d of 2.5% DSS (n = 3 replicates/group, except n = 5 for the water-treated WT and n = 4 for the 7-d post-DSS WT). Scale bars, 50 $\mu$m. All experiments in (C, D, E) were n = 3 independent experiments. In (E), data are shown as box-and-whisker plots. The boxes represent the interquartile range, the whiskers represent 1.5 * interquartile range, and the individual point represents an outlier. NS, not significant. * denotes $P < 0.05$. NES = normalized enrichment score, FDR-q = false discovery rate q-value.

concentration was determined using Bicinchoninic Acid Protein Assay Kit.

### ELISA for CRP

Six micrograms of the protein extract was used for the CRP assay using a CRP assay kit. The flash-frozen protein extracts, after normalizing the concentration, were used for the CRP assay. Standards were prepared freshly, and the CRP assay was carried out as per the manufacturer's recommendation.

### Histology, immunostaining, and image acquisition

The freshly isolated colon was flushed with PBS, opened longitudinally, made into Swiss rolls, and fixed overnight at 4°C with 4% PFA in PBS. The fixed tissues were dehydrated in alcohol series and processed in xylene and paraffin before embedding them in paraffin. 5-$\mu$m sections of the paraffin-embedded tissue were used for histological evaluation and immunostaining as previously described (86). A summary of the antibodies and dilution used is summarized in Tables S1, S2, and S3. Hematoxylin or methyl green was used as a nuclear counterstain. Brightfield images were obtained using a Nikon microscope (Model Eclipse Ci-L; cat. No. M568E) and a Nikon camera (DS-Fi3; cat. No. 117837). Fluorescent images were obtained using a Zeiss confocal microscope (LSM 880; Carl Zeiss Microscopy GmbH). The same laser power and gain were maintained when obtaining fluorescent images. Low-magnification (4X) fluorescent images were obtained using a BioTek Lionheart FX microscope and camera. Contrast and brightness, when adjusted for brightfield images, were uniformly applied across the treatments being compared.

### BrdU–EdU pulse-chase and fluorescent detection

Mice were injected with 1 mg of EdU and 1 mg of BrdU at one and 6 h, respectively, before euthanizing the mice. Fluorescent immunohistochemistry was used to detect the BrdU-incorporated cells; the EdU-incorporated cells within the same tissue were codetected with a red fluorescent marker as per the manufacturer's recommendation. The distance of migration was calculated by measuring the distance between the top BrdU-positive and the EdU-positive cell relative to the height of the crypt being accounted. The quantification was performed on the images captured from the distal colon. At least 30 open crypts were accounted for each sample.

### Picrosirius red staining and assessment of collagen proportionate area

Five-micron sections of paraffin-embedded colons were hydrated and stained for collagen using a Sirius Red/Fast Green kit as per the manufacturer's recommendation. Collagen deposition was assessed only in the mucosae; that is, collagen in the submucosa, mucosa, muscularis, and serosa were not accounted. Collagen-positive pixels were quantified as described earlier (34). Briefly, five 10x images from the distal colon were randomly selected for the assessment. The images were then deconvoluted to visualize only the collagen-positive pixels and quantified to obtain collagen-positive pixels per unit area of the mucosa.

### Quantification of intact crypts and crypt-less mucosal regions

Ten randomly selected 20x images of $\alpha$-SMA immunostaining in the distal colon were used. The number of intact crypts per unit area only in the mucosa was counted manually. The same images were used to quantify the crypt-less regions per unit area of the mucosa, containing $\alpha$-SMA–positive and $\alpha$-SMA–negative cells.

### Quantification of Col1a1 (type I collagen) in the pericryptal epithelium

Ten 20x images of the distal colon immunostained for Col1a1 for each replicate were used. Col1a1 immunoreactivity in the pericryptal epithelium of at least 50 open crypts for each replicate was quantified as follows (Fig S2G): a rectangle was drawn around the pericryptal epithelium to assign the total area accounted for each crypt. Using ImageJ 1.53t and Zen 3.1 software, the Col1a1 immunoreactive region in the pericryptal epithelium was outlined. The outlined Col1a1-positive area per unit area of the rectangle was quantified.

### Quantification of p-H3 (Ser10)–labeled cells to assess proliferation

The p-H3 (Ser10) (histone H3 [p Ser10])–positive cells were detected by p-H3 (Ser10) immunohistochemistry. p-H3 (Ser10)–positive cells in at least 40 open and intact crypts per replicate of the distal colon were enumerated.

### Statistical analysis

Data visualization was performed in GraphPad Prism 8.0.2 for bar graphs or in R (v 4.3.1) packages, ggplot2 (v 3.5.1) and ggprism (v 1.0.4), for a box-and-whisker plot. In all bar graphs, data are means ± SEM from three independent experiments. Unpaired $t$ tests were used to compare the means of data between two groups. Statistical significance was set at $P \leq 0.05$. For the gene count data, $P$adj. values obtained from DEG analysis were used to determine the statistical significance. The statistical details for each graph, including the number of independent experiments and statistical significance, are indicated in the figure and/or figure legends.

### RNA-seq library preparation and sequencing

Quality control, library preparation, and sequencing for the study were conducted by Novogene Corporation Inc. RNA integrity was initially assessed using RNA Nano 6000 Assay Kit of the Bioanalyzer 2100 system (Agilent Technologies), and only RNA samples with an RIN value above 4 were selected for further processing. This process involved the purification of mRNA and subsequent cDNA synthesis. After adapter ligation, the cDNA underwent another purification step. The prepared library's quality was then evaluated using the Agilent Bioanalyzer 2100 system. Finally, sequencing was performed on an Illumina NovaSeq platform, generating 150-bp paired-end reads. The raw sequencing reads, provided in a FASTQ format, were processed by Novogene using their in-house Perl scripts to obtain clean reads. These clean reads were then aligned to the mm9 mouse reference genome using Hisat2 (v2.0.5) (87). Subsequently,

the aligned reads were quantified for gene expression levels using featureCounts (v1.5.0-p3) (88).

## Bioinformatics

Differential expression analysis for pairwise comparisons was performed using the DESeq2 (v1.40.2) package (89) in R (v4.3.1) (90). *P*-values were adjusted using the Benjamini–Hochberg method to control the false discovery rate (91). Genes with an adjusted *P*-value (*P*adj.) less than 0.05 were considered significantly differentially expressed.

Gene Ontology (GO) (92) enrichment analysis was performed on the significantly DEGs using the clusterProfiler (v4.9.4) R package (93). *P*-values obtained from the hypergeometric test were adjusted using the Benjamini–Hochberg method for multiple hypothesis testing. GO terms with an adjusted *P*-value (*P*adj) less than 0.05 were deemed significantly enriched.

Normalized counts for GSEA were obtained using the counts (dds, normalized = TRUE) function from DESeq2, and GSEA (v4.3.2) was conducted using the Broad Desktop Application (94, 95). Mouse Ensembl IDs were mapped to human gene symbols, and genes were ranked using the Signal2Noise metric. The analysis used the weighted enrichment statistic and 10,000 gene set permutations, with gene sets filtered by size (minimum 15, maximum 500). Gene sets with a false discovery rate q-value below 0.05 were identified as significantly enriched.

Gene sets were sourced from the Molecular Signatures Database (MSigDB) (96). Heatmaps of core-enriched genes and UpSet plots were generated using the ComplexHeatmap package in R (97). UpSet plots were used to determine the uniquely enriched gene sets across each pairwise comparison that was performed.

Ingenuity Pathway Analysis (IPA) (98) was performed with the DEGs identified in our study. DEGs with $|Log_2FoldChange| \geq 0.584$ were uploaded to IPA for functional annotation and regulatory network analysis.

## Data Availability

Normalized transcript count data have been submitted to the National Center for Biotechnology Information Gene Expression Omnibus, GSE252864, and raw sequences to the Sequence Read Archive. The code is archived on Zenodo: https://zenodo.org/doi/10.5281/zenodo.13772322.

## Supplementary Information

## Acknowledgements

A Perekatt was supported by K22ACTFCA218462. MP Verzi was supported by NIDDK126446. C Lin was supported by NSF REU site award #2050921.

## Author Contributions

Z Hashemi: data curation, formal analysis, supervision, validation, investigation, visualization, methodology, project administration, and writing—review and editing.

T Hui: data curation, formal analysis, supervision, validation, investigation, visualization, methodology, project administration, and writing—review and editing.

A Wu: data curation, formal analysis, validation, investigation, visualization, and methodology.

D Matouba: data curation, formal analysis, investigation, visualization, and methodology.

S Zukowski: data curation, software, formal analysis, visualization, and project administration.

S Nejati: formal analysis, validation, investigation, visualization, and methodology.

C Lim: software and formal analysis.

J Bruzzese: formal analysis, visualization, and methodology.

C Lin: formal analysis and investigation.

K Seabold: formal analysis and investigation.

C Mills: formal analysis, validation, and investigation.

K Wrath: validation and methodology.

H Wang: resources and methodology.

H Wang: conceptualization, resources, review, and editing

MP Verzi: conceptualization, resources, supervision, funding acquisition, validation, investigation, methodology, and project administration.

A Perekatt: conceptualization, resources, formal analysis, supervision, funding acquisition, validation, investigation, methodology, project administration, and writing—original draft, review, and editing.

## Conflict of Interest Statement

The authors declare that they have no conflict of interest.

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
