## [Reviewer comments · Life Science Alliance]

Life Science Alliance

Epithelial-specific Loss of Smad4 Alleviates the Fibrotic Response in an Acute Colitis Mouse Model.

Zahra Hashemi, Thompson Hui, Alex Wu, Dahlia Matouba, Steven Zukowski, Shima Nejati, Crystal Lim, Julianna Bruzzese, Cindy Lin, Kyle Seabold, Connor Mills, Kylee Wrath, Haoyu Wang, Hongjun Wang, Michael P Verzi, Ansu Perekatt
DOI: <https://doi.org/10.26508/lsa.202402935>

Corresponding author(s): Dr. Ansu Perekatt (Stevens Institute of Technology)

Review Timeline:

Submission Date:	2024-07-09
Editorial Decision:	2024-09-04
Revision Received:	2024-09-24
Editorial Decision:	2024-09-25
Revision Received:	2024-09-27
Accepted:	2024-09-27

Transaction Report:

September 4, 2024

Re: Life Science Alliance manuscript #LSA-2024-02935-T

Dr. Ansu Perekatt
Stevens Institute of Technology
Chemistry and Chemical Biology
507 River St.
McLean 206
Hoboken, NJ 07030

Dear Dr. Perekatt,

Thank you for submitting your manuscript entitled "Epithelial-specific loss of Smad4 in the Colon Enhances the Wound Healing Response and Alleviates the Fibrotic Response in an Acute DSS Mouse Model" to Life Science Alliance. The manuscript was assessed by expert reviewers, whose comments are appended to this letter. We invite you to submit a revised manuscript addressing the Reviewer comments.

Thank you for this interesting contribution to Life Science Alliance. We are looking forward to receiving your revised manuscript.

Sincerely,

B. MANUSCRIPT ORGANIZATION AND FORMATTING:

Reviewer #1 (Comments to the Authors (Required)):

Summary

In this study Smad4 was knocked out using a tamoxifen inducible Villin-cre recombinase to examine the effects of epithelial specific loss of Smad4 on response to acute exposure to DSS. This study demonstrates that acute 3- and 7-day DSS treatment causes epithelial damage and changes associated with fibrosis in the WT mouse, and that short term loss of Smad4 alleviates these changes. Key findings in the WT mouse include colitis, decreased proliferation, increased collagen deposition and enrichment of inflammatory gene signatures. In the KO mouse key findings include limitation of morphological damage, expansion of the proliferative zone, enrichment of regenerative gene signatures, increased expression of collagens associated with wound healing, and negative enrichment of proinflammatory gene signatures. By using an acute DSS exposure and short term epithelial specific knockout of Smad4 this study was able to examine early, more direct effects of loss of Smad4 on intestinal epithelial damage. This study demonstrates a role for the epithelium in protection from intestinal fibrosis and shows that in this specific context loss of Smad4 activates a range of processes that are protective against epithelial damage and fibrosis.

Are data supportive?

Overall data are strongly supportive of the main points of the paper.

1. "Epithelial-specific loss of Smad4 in the acute DSS mouse model alleviates the colitis response". The data clearly support the conclusion.
2. "Epithelial-specific loss of Smad4 promotes the regenerative response in the epithelium."
Fig. 1A and 1B show that proliferation is maintained but not increased in the KO and GSEA analysis (Fig. 1F) shows enrichment of a regenerative gene signature in both treated and untreated KO, thus demonstrating enhanced regenerative response in untreated KO relative to WT. In addition, the BrdU and EdU experiments show expansion of the proliferative zone in the DSS treated KO which indicates an enhanced response in the DSS treated vs untreated KO mouse.
3. "Epithelial-specific loss of Smad4 alleviates the fibrotic response in the acute DSS mouse model."
Fig. 3A and 3B clearly demonstrate increased collagen deposition in the WT. Fig. 3C shows peri-cryptal expression of alpha-SMA which parallels peri-cryptal expression of protective Col1a1 in Fig. 4D and so is also supportive. However, it is unclear how to interpret Fig. 3D It is not clear if Fig. 3D is meant to compare the extent of loss of crypts within the mucosa or the extent of alpha-SMA-positive areas in the crypt-less mucosal regions.
4. "The epithelial transcriptome in the 3-day DSS-treated in the SMAD4 IEC-KO supports the wound healing response."
Fig. 4C, 4D and 4E which show increased expression of collagens implicated in wound healing and peri-cryptal localization of Col1a1 in the KO are supportive.
However, in Fig. 4A and B it is not clear which of the processes and genes shown to be enriched in the KO promote wound healing and which do not. Given that collagens are involved in promoting both fibrosis and regeneration it would be helpful to clarify.
5. "Epithelial-specific loss of Smad4 attenuates the DSS-induced inflammatory response in the acute DSS mouse model."
The data clearly support the conclusion.

Additional comments

Given the complexity of the roles of Smad4 in the colon epithelium and the complexity of epithelial regeneration, it would be helpful to see further discussion of the results of this study in the context of previous work.

1. With respect to the regenerative response, Figures 2A, C-E, and Supp. Fig. S2E indicate an expansion of the proliferative zone. A potential cause of this expansion is altered crypt cell type composition, in particular impaired differentiation and increased stem cell activity. Another study of epithelial regeneration has identified changes in stem cell composition during the

repair process (PMID: 31708126). It would be helpful to see comments on the potential impact of Smad4 KO on crypt cell type composition, especially stem and progenitor cells.

2. The current study shows enhanced epithelial regeneration and decreased fibrotic changes after DSS epithelial damage in the Smad4 KO but a previous study (PMID: 37865088) showed that TGF beta promotes epithelial repair in a model of irradiation-induced colon epithelial damage. Comparing these two responses could give insight into the specific factors that influence epithelial repair in different contexts. It would be helpful to see comments on possible causes for these two different responses.

Reviewer #2 (Comments to the Authors (Required)):

Authors of this manuscript studied the epithelial-specific loss of Smad4 in the mouse colon and its effects on wound healing response by using acute Dextran Sulphate Sodium inflammatory bowel disease mouse model. They have shown epithelial-specific loss of Smad4 is associated with alleviated fibrotic response and promoted mucosal healing.

Remarks

1. Please discuss how these results could be translated into clinic, having in mind Smad4 loss is associated with increased risk of colon cancer in DSS mouse model
2. What would have happened if the mice were grown for longer periods of time, would cancer arise at a later time point?
3. Please explain why were time points of 3 days and 7 days selected. What is expected to be seen at these time points phenotypically?
4. Please add legends for Figure 2D and 2E.

P.S. There is a preprint of this manuscript on bioRxiv
<https://doi.org/10.1101/2024.03.08.578000>

The revisions in the manuscript are in red fonts, and responses to the reviewers' comments are in blue fonts.

Reviewer #1 (Comments to the Authors (Required)):

Summary

In this study Smad4 was knocked out using a tamoxifen inducible Villin-cre recombinase to examine the effects of epithelial specific loss of Smad4 on response to acute exposure to DSS. This study demonstrates that acute 3- and 7-day DSS treatment causes epithelial damage and changes associated with fibrosis in the WT mouse, and that short term loss of Smad4 alleviates these changes. Key findings in the WT mouse include colitis, decreased proliferation, increased collagen deposition and enrichment of inflammatory gene signatures. In the KO mouse key findings include limitation of morphological damage, expansion of the proliferative zone, enrichment of regenerative gene signatures, increased expression of collagens associated with wound healing, and negative enrichment of proinflammatory gene signatures. By using an acute DSS exposure and short term epithelial specific knockout of Smad4 this study was able to examine early, more direct effects of loss of Smad4 on intestinal epithelial damage. This study demonstrates a role for the epithelium in protection from intestinal fibrosis and shows that in this specific context loss of Smad4 activates a range of processes that are protective against epithelial damage and fibrosis.

Are data supportive?

Overall data are strongly supportive of the main points of the paper.

Dear Reviewer,

We are incredibly grateful for the review, suggestions, and comments. By addressing the comments, we believe the manuscript has improved greatly.

Please see the response to the comments.

1. "Epithelial-specific loss of Smad4 in the acute DSS mouse model alleviates the colitis response". The data clearly support the conclusion.

We are grateful for the comment.

2. "Epithelial-specific loss of Smad4 promotes the regenerative response in the epithelium."

Fig. 1A and 1B show that proliferation is maintained but not increased in the KO and GSEA analysis (Fig. 1F) shows enrichment of a regenerative gene signature in both treated and untreated KO, thus demonstrating enhanced regenerative response in untreated KO relative to WT. In addition, the BrdU and EdU experiments show expansion of the proliferative zone in the DSS treated KO which indicates an enhanced response in the DSS treated vs untreated KO mouse.

Thank you for the comprehensive assessment.

3. "Epithelial-specific loss of Smad4 alleviates the fibrotic response in the acute DSS mouse model."

Fig. 3A and 3B clearly demonstrate increased collagen deposition in the WT. Fig. 3C

shows peri-cryptal expression of alpha-SMA which parallels peri-cryptal expression of protective Col1a1 in Fig. 4D and so is also supportive. However, it is unclear how to interpret Fig. 3D. It is not clear if Fig. 3D is meant to compare the extent of loss of crypts within the mucosa or the extent of alpha-SMA-positive areas in the crypt-less mucosal regions.

Answer: We apologize for the confusion. Figure 3D compares the extent of crypt loss due to the replacement of the epithelia by stroma, which also contains alpha-SMA-positive cells. The figure legend in Figure 3D has been modified to “**The relative proportion of the mucosa devoid of intact epithelial crypts**” to avoid confusion.

Figure 3D:

4. "The epithelial transcriptome in the 3-day DSS-treated in the SMAD4 IEC-KO supports the wound healing response."

Fig. 4C, 4D and 4E which show increased expression of collagens implicated in wound healing and peri-cryptal localization of Col1a1 in the KO are supportive.

However, in Fig. 4A and B it is not clear which of the processes and genes shown to be enriched in the KO promote wound healing and which do not. Given that collagens are involved in promoting both fibrosis and regeneration it would be helpful to clarify.

Thank you for this suggestion, especially since collagens are implicated in both fibrosis and epithelial homeostasis.

Answer: Figure 4A indicates that Smad4-loss-induced transcriptional changes support epithelial-specific ECM organization and wound healing. Several of the genes in the ECM constituent signature have been implicated in wound healing: *Ltbp1* (PMID: 35456902), *PRG4* (PMID: 35750773), *DCN* (PMID: 7529785). Collagens have been implicated in both fibrosis and wound healing. However, the fibrotic versus wound healing attributes depend on the source of collagen and the site of its deposition. For example, fibroblast deposition of type I and III collagens in the mesenchyme is attributed to fibrosis (reviewed in PMID: 35931028). At the same time, the same collagens in the epithelial basement membrane are associated with wound healing-associated processes such as proliferation and differentiation. Type IV collagen in the epithelial basement membrane is implicated in the tissue integrity (PMID: 28736303). While increased type VI collagens are detected in intestinal strictures of CD patients (PMID: 30452921), epithelial BM-associated type VI collagen is known to promote cell

spreading and wound closure in the lung epithelium (PMID: 30550606) and modulates epithelial migration of the intestinal epithelium {PMID: 21406227}.

To address this point, we've made the following edits in the manuscript:

To the last paragraph on Page 9:

Several of the genes in the ECM constituent signature, such as *Ltbp1*(40), *Prg4*(41), and *Dcn*(42), have been implicated in wound healing. Interestingly, members of small Leucine Rich Repeat Proteoglycans (SLRPs), which are downregulated during pathological progression of colitis (SLRPs)(43), were positively enriched in the ECM signature, suggesting protective effects of the Smad4^{IEC-KO} ECM constituents.

Additionally, the transcript levels of genes encoding the collagens implicated in wound healing(44) such type I, II, IV and VI(29) were higher in the Smad4 knockout colon epithelium (Fig 4C).

To the last paragraph on Page 13:

The fibrosis versus the wound healing attributes of various collagens depends on the source of collagen and the site of its deposition. For example, fibroblast deposition of type I and III collagens in the mesenchyme is attributed to fibrosis (reviewed in PMID: 35931028), whereas type I and III collagens, when present in the epithelial ECM are associated with wound healing (PMID: 36794945) Likewise, while type VI collagen in epithelial basement membrane is associated with wound healing processes (PMID: 30550606) (PMID: 21406227), an increase in type VI collagen has been reported in the strictures of CD, and collagenous colitis UC show (PMID: 30452921).

Type I collagen in the epithelial ECM promotes epithelial proliferation(46), migration, and differentiation even when cell-cell contact is absent(45,76). Since the denuded epithelium resembles epithelial cells lacking cell-cell contact, the increased type I collagen in the pericryptal region (Fig 4D and E) might restore faster homeostasis.

5. "Epithelial-specific loss of Smad4 attenuates the DSS-induced inflammatory response in the acute DSS mouse model."

The data clearly support the conclusion.

Additional comments

Given the complexity of the roles of Smad4 in the colon epithelium and the complexity of epithelial regeneration, it would be helpful to see further discussion of the results of this study in the context of previous work.

1. With respect to the regenerative response, Figures 2A, C-E, and Supp. Fig. S2E indicate an expansion of the proliferative zone. A potential cause of this expansion is altered crypt cell type composition, in particular impaired differentiation and increased stem cell activity. Another study of epithelial regeneration has identified changes in stem cell composition during the repair process (PMID: 31708126). It would be helpful to see comments on the potential impact of Smad4 KO on crypt cell type composition, especially stem and progenitor cells.

Answer: The *Lgr5* transcripts levels were significantly higher in the untreated and the DSS-treated *Smad4* KO than its wild type counterpart, indicating increased stem cell activity in the DSS-treated *Smad4* knockout (Fig. S3F). This observation is consistent with the previous report on the *Lgr5* cells in irradiation-induced intestinal epithelial regeneration (PMID: 26503053). However, the gene expression changes showed no evidence of an altered proportion of colitis-associated regenerative stem cells (CARSCs) reported in the 2D culture model mimicking the homeostasis-injury-regeneration model (PMID: 31708126). This difference could be due to the acute nature of the colitis model, differences between the *in vivo* versus *in vitro* model, and the inability to capture the hypoxic switch possible in the 2D model. Additionally, we are yet to perform immunoassays to evaluate the differences in the relative proportion of the various cell types, if any, that were not captured through gene expression profiling. To address this point, we've made the following edits in the text. the first paragraph on page 12:

Additionally, *Lgr5* transcript levels were significantly higher in the DSS-treated *Smad4*^{IEC-KO} colon epithelium (Supplementary. Fig S2H), consistent with the role of *Lgr5*⁺ stem cells in the intestinal epithelial regeneration (PMID: 28059064).

Supplementary Figure S3H:

2. The current study shows enhanced epithelial regeneration and decreased fibrotic changes after DSS epithelial damage in the *Smad4* KO but a previous study (PMID: 37865088) showed that TGF beta promotes epithelial repair in a model of irradiation-induced colon epithelial damage. Comparing these two responses could give insight into the specific factors that influence epithelial repair in different contexts. It would be helpful to see comments on possible causes for these two different responses.

Answer: This is an intriguing point. Distinct modes of epithelial loss precede the regenerative response to irradiation versus DSS. While irradiation triggers DNA-damage-induced intrinsic apoptosis (PMID: 10602483), DSS causes epithelial breaches that expose the epithelium and sub-epithelial region to luminal contents. As the apoptotic cells do not release their immunogenic intracellular contents, but are ingested by resident phagocytes (PMID: 18039143), the inflammatory response in intrinsic DNA-damage-induced apoptosis is minimal. However, exposure to the luminal content following DSS-induced epithelial breach triggers inflammatory signaling through the engagement of damage-associated molecular patterns (DAMPs) on the epithelial cells, which, by engaging the pattern recognition receptors (PRRs) on the immune cells

(PMID: 26931062) might lead to the various lytic forms of cell death, which in turn might prolong the inflammatory response.

TGF-beta has context-dependent functions. TGF-beta promotes epithelial regeneration following IR-induced apoptosis and fibrosis during chronic inflammation. While, after IR-induced apoptosis, TGF-beta-induced fetal reprogramming in the epithelium (PMID: 37865088) and TGF-beta-induced suppression of inflammation promotes epithelial regeneration (PMID: 11781349), TGF-beta promotes stromal cell proliferation and ECM deposition that promote fibrosis (PMID: 15117886) during chronic inflammation. In addition, Smad4 independent-TGF signaling targets might also be involved in epithelial regeneration, the plausibility and targets of which are currently being explored.

To address this point, we have made the following edits.

On page 13 in the discussion section, we have added the following paragraph.

Given that Smad4 is a transcriptional effector TGF- β signaling, our finding that Smad4 loss promotes epithelial regeneration is intriguing. TGF- β promotes epithelial regeneration after ionizing radiation (IR) through fetal reprogramming (PMID: 37865088), and by suppressing inflammation (PMID: 11781349). However, the inflammatory response is minimal after IR compared to DSS. The subdued inflammatory response following IR is attributed to the phagocytotic clearance of the apoptotic cells (PMID: 10602483) (PMID: 18039143). On the other hand, DSS-induced epithelial breach exposes the luminal contents to the epithelium, thereby engaging damage-associated molecular patterns (DAMPs) on the epithelium (PMID: 26931062), which might trigger various lytic forms of cell death, which in turn might prolong the inflammatory response.

Although the Smad4^{IEC-KO} colon showed no increase in mesenchymal collagen deposition in the mucosa after DSS (Fig 3A and B), the pericryptal collagen deposition was higher (Fig 4D) in the Samd4^{IEC-KO} colon.

Reviewer #2 (Comments to the Authors (Required)):

Authors of this manuscript studied the epithelial-specific loss of Smad4 in the mouse colon and its effects on wound healing response by using acute Dextran Sulphate Sodium inflammatory bowel disease mouse model. They have shown epithelial-specific loss of Smad4 is associated with alleviated fibrotic response and promoted mucosal healing.

Dear Reviewer,

We are incredibly grateful for the review, suggestions, and comments. By addressing the comments, we believe the manuscript has improved greatly.

Please see the response to the comments.

Remarks

1. Please discuss how these results could be translated into clinic, having in mind Smad4 loss is associated with increased risk of colon cancer in DSS mouse model

Answer: Our study reveals enhanced regenerative response in the Smad4-KO is associated with epithelial-ECM changes and alleviated fibrosis. Hence, exploiting the epithelial-ECM changes that promotes epithelial regeneration is a potential strategy against fibrosis.

To address this point, we have added the following last paragraph to the discussion on page 15:

In conclusion, our study reveals that enhanced regenerative response in the Smad4^{IEC-KO} is associated with epithelial-ECM changes and alleviated fibrosis. Hence, exploiting the epithelial-ECM changes that promote epithelial regeneration is a potential strategy against fibrosis in IBDs.

2. What would have happened if the mice were grown for longer periods of time, would cancer arise at a later time point?

Answer: Yes. Previous literature has shown that Smad4 knock out mice develop tumors in the presence of DNA-damaging agents with (PMID: 30109253) or without DSS treatment (PMID: 29986996). Furthermore, the immunosuppressive milieu in the DSS-treated Smad4 knockout (Fig 5 C&D) mice expected to enhance tumorigenesis by attenuating the immune cells activities that direct tumor cells.

To address this point, the following text has been added in the first paragraph on page 12.

However, given the role of Smad4 in genomic stability(59,60,61) and tumor suppression(62,63), tumorigenesis in the Smad4 knockout chronic DSS mouse model is not surprising, especially in the presence of a DNA-damaging agent such as AOM (Azoxymethane)(55,56,57). **Therefore, we expect tumorigenesis in the DSS-treated Smad4^{IEC-KO} colon, especially considering the immunosuppressive milieu (Fig 5C and D) if a long-term DSS-regimen for chronic colitis was adopted.**

3. Please explain why were time points of 3 days and 7 days selected. What is expected to be seen at these time points phenotypically?

Answer: We chose the DSS-induced loss of colonic epithelium is minimal at the 3-day time point, enabling collection of the sufficient quality of epithelial tissue for transcriptomic analysis at a time-point when early molecular responses are evident (PMID: 20467900). Since the gross phenotypic change manifests within seven days of continuous DSS treatment in acute colitis model, we chose the 7-day timepoint to evaluate the gross phenotypic effects of DSS after 7 days of 2.5% DSS.

To address this point, the following text has been added in the introduction section on page 4.

Smad4 was knocked out specifically in the intestinal epithelium (Smad4^{IEC-KO}), followed by DSS treatment. Epithelial-specific transcriptional and molecular changes were

assessed after three days of 2.5% DSS (3-day post-DSS). We chose the three-day time point as the DSS-induced loss of the colonic epithelial tissue is minimal at this time point – enabling collection of the sufficient quality of epithelial tissue for transcriptomic analysis at a timepoint when early molecular responses are evident (PMID: 20467900). Since the gross phenotypic change manifests within seven days of continuous DSS treatment in acute colitis model (PMID: 24510619), we chose the seven-day time point

4. Please add legends for Figure 2D and 2E.

Answer: The legends have been added

September 25, 2024

RE: Life Science Alliance Manuscript #LSA-2024-02935-TR

Dr. Ansu Perekatt
Stevens Institute of Technology
Chemistry and Chemical Biology
507 River St.
McLean 206
Hoboken, NJ 07030

Dear Dr. Perekatt,

Thank you for submitting your revised manuscript entitled "Epithelial-specific Loss of Smad4 Alleviates the Fibrotic Response in an Acute Colitis Mouse Model.". We would be happy to publish your paper in Life Science Alliance pending final revisions necessary to meet our formatting guidelines.

- please be sure that the authorship listing and order is correct
- please make sure that all author names are correct in the manuscript (there's a discrepancy with the spelling of one of the authors' last names in the system and the manuscript)
- please upload your tables as editable doc or excel files

A. FINAL FILES:

B. MANUSCRIPT ORGANIZATION AND FORMATTING:

**Submission of a paper that does not conform to Life Science Alliance guidelines will delay the acceptance of your

manuscript.**

The license to publish form must be signed before your manuscript can be sent to production. A link to the electronic license to publish form will be available to the corresponding author only. Please take a moment to check your funder requirements.

Thank you for your attention to these final processing requirements. Please revise and format the manuscript and upload materials within 4 days.

Sincerely,

September 27, 2024

RE: Life Science Alliance Manuscript #LSA-2024-02935-TRR

Dr. Ansu Perekatt
Stevens Institute of Technology
Chemistry and Chemical Biology
507 River St.
McLean 206
Hoboken, NJ 07030

Dear Dr. Perekatt,

Thank you for submitting your Research Article entitled "Epithelial-specific Loss of Smad4 Alleviates the Fibrotic Response in an Acute Colitis Mouse Model.". It is a pleasure to let you know that your manuscript is now accepted for publication in Life Science Alliance. Congratulations on this interesting work.

DISTRIBUTION OF MATERIALS:

Again, congratulations on a very nice paper. I hope you found the review process to be constructive and are pleased with how the manuscript was handled editorially. We look forward to future exciting submissions from your lab.

Sincerely,
